# Intraoperative Visualization and Treatment of Salivary Gland Dysfunction in Sjögren’s Syndrome Patients Using Contrast-Enhanced Ultrasound Sialendoscopy (CEUSS)

**DOI:** 10.3390/jcm12124152

**Published:** 2023-06-20

**Authors:** K. Hakki Karagozoglu, Anissa Mahraoui, Joseph C. J. Bot, Seunghee Cha, Jean-Pierre T. F. Ho, Marco N. Helder, Henk S. Brand, Imke H. Bartelink, Arjan Vissink, Gary A. Weisman, Derk Hendrik Jan Jager

**Affiliations:** 1Department of Oral and Maxillofacial Surgery/Oral Pathology, Amsterdam UMC, Location Vrije Universiteit Amsterdam, De Boelelaan 1118, 1081 HV Amsterdam, Noord-Holland, The Netherlands; 2Department of Radiology and Nuclear Medicine, Amsterdam UMC, Location Vrije Universiteit Amsterdam, De Boelelaan 1117, 1081 HV Amsterdam, Noord-Holland, The Netherlands; 3Department of Oral and Maxillofacial Diagnostic Sciences, Center for Orphaned Autoimmune Disorders, University of Florida, 1395 Center Drive, Gainesville, FL 32610, USA; 4Department of Oral Biochemistry, Academisch Centrum Tandheelkunde Amsterdam, Gustav Mahlerlaan 3004, 1081 LA Amsterdam, Noord-Holland, The Netherlands; 5Department of Pharmacy, Amsterdam UMC, Location Vrije Universiteit Amsterdam, De Boelelaan 1117, 1018 HV Amsterdam, Noord-Holland, The Netherlands; 6Department of Oral and Maxillofacial Surgery, University Medical Center Groningen, University of Groningen, Hanzeplein 1, 9713 GZ Groningen, Groningen, The Netherlands; 7Department of Biochemistry, Christopher S. Bond Life Sciences Center, University of Missouri, 1201 Rollins St, Columbia, MO 65211, USA; 8Amsterdam Institute for Infection and Immunity, Inflammatory Diseases, De Boelelaan 1118, 1081 HV Amsterdam, Noord-Holland, The Netherlands

**Keywords:** saliva, Sjogren’s syndrome, Sjögren’s syndrome, sialendoscopy, endoscopy, ultrasound, microbubbles, xerostomia

## Abstract

In sialendoscopy, ducts are dilated and the salivary glands are irrigated with saline. Contrast-enhanced ultrasound sialendoscopy (CEUSS), using microbubbles, may facilitate the monitoring of irrigation solution penetration in the ductal system and parenchyma. It is imperative to test CEUSS for its safety and feasibility in Sjögren’s syndrome (SS) patients. CEUSS was performed on 10 SS patients. The primary outcomes were safety, determined by the occurrence of (serious) adverse events ((S)AEs), and feasibility. The secondary outcomes were unstimulated and stimulated whole saliva (UWS and SWS) flow rates, xerostomia inventory (XI), clinical oral dryness score, pain, EULAR Sjögren’s syndrome patient reported index (ESSPRI), and gland topographical alterations. CEUSS was technically feasible in all patients. Neither SAEs nor systemic reactions related to the procedure were observed. The main AEs were postoperative pain (two patients) and swelling (two patients). Eight weeks after CEUSS, the median UWS and SWS flow had increased significantly from 0.10 to 0.22 mL/min (*p* = 0.028) and 0.41 to 0.61 mL/min (*p* = 0.047), respectively. Sixteen weeks after CEUSS, the mean XI was reduced from 45.2 to 34.2 (*p* = 0.02). We conclude that CEUSS is a safe and feasible treatment for SS patients. It has the potential to increase salivary secretion and reduce xerostomia, but this needs further investigation.

## 1. Introduction

Sjögren’s syndrome (SS) is a connective tissue disease in which patients suffer from inflammation of the exocrine glands, in particular, the tear and salivary glands, resulting in dry mouth and dry eyes. In addition, people with SS often suffer from extraglandular manifestations, including fatigue and joint pain [1,2]. SS is considered the most common rheumatoid autoimmune disorder following rheumatoid arthritis (RA) and predominantly affects women, with a 9:1 female:male ratio [3,4]. In many people, SS often occurs as an isolated condition, but it can also co-occur with other rheumatic conditions, such as RA and systemic lupus erythematosus (SLE). The pathogenesis of SS is still largely unknown. However, it has become clear that disturbances of the immune system play a central role in the etiology, but whether this is a primary cause of SS or a consequence of a previous (viral) infection or other extrinsic factors remains uncertain [4].

SS is characterized by mononuclear infiltrates and IgG-producing plasma cells in the salivary and lacrimal glands. This infiltration leads to the destruction of glandular tissues, resulting in decreased salivary secretion and leading to hyposalivation [5]. Hyposalivation is associated with an increased risk of developing yeast and bacterial infections, speech and digestive disorders, burning sensations, loss of taste, and dental caries, all of which severely reduce the quality of life of SS patients [1]. Additionally, patients sometimes suffer from painful episodes of sialadenitis with partial or complete obstruction of the saliva-transporting ducts. These obstructions are presumed to be caused by intraductal formations of scar tissue and fibrinous plaques [6,7,8].

Treatments for xerostomia related to SS are limited to the use of external hydration; gustatory, electronic, or mechanical stimulants; and muscarinic receptor agonists that induce salivary secretion from residual acinar cells, and saliva substitutes. Such remedies are considered inadequate for most patients with SS, so the development of more effective SS treatments is essential [9,10,11]. Treatment with antirheumatic drugs was shown to be effective in some SS patients. However, this systemic approach causes side effects, has minimal impact on salivary gland function, and is effective only in subgroups of SS patients [12,13]. Thus, other approaches are necessary to improve saliva flow and prevent salivary gland degeneration in SS patients.

In recent years, we have investigated a promising sialendoscopic treatment for patients with SS. In this technique, the ducts of the salivary glands are irrigated with saline or steroids [14,15]. This minimally invasive technique allows the dilatation of ductal strictures and removal of ductal debris. We, and other groups, showed that this approach alleviated some of the oral symptoms of patients suffering from SS and partially restored glandular function (i.e., saliva flow rate) [14,15,16,17,18,19]. Although sialendoscopy in SS patients has shown promising results, there are several limitations in the current approach. It is not possible to enter and explore all ducts with the sialendoscope because of decreasing ductal diameters from the main duct to terminal glandular tissue. In addition, it is difficult to intraoperatively monitor the delivery and degree of penetration of the irrigation fluids or medications into the ductal system and parenchyma, as well as to evaluate therapeutic efficacy (i.e., removal of obstructions and restoration of flow rate) [20].

To address these limitations of sialendoscopy and improve our therapy, we postulated that contrast-enhanced ultrasound sialendoscopy (CEUSS), using commercially available sulfur hexafluoride microbubbles (SonoVue; Bracco Imaging SpA, Milan, Italy), might be useful [20]. These microbubbles, consisting of 5–10 µm gas-filled particles, are currently used in the cardiology field to image and dissolve arterial occlusions (sonothrombolysis) [21,22,23,24]. In CEUSS, these microbubbles are postulated to aid non-invasive high-resolution ultrasound (US) imaging of the glands and visualize ductal obstructions (where microbubbles should accumulate and not penetrate the gland). Moreover, the dissolution of ductal obstructions by hydrostatic pressure could be observed in real time. Thus, CEUSS should have value over normal sialendoscopy since it encompasses real-time non-invasive intra-operative visualization and diagnostic imaging.

Microbubbles were already utilized for US-assisted contrast imaging in salivary glands [25,26], but not for US-assisted contrast imaging in salivary glands of patients affected by SS. Therefore, because it concerns an intervention with a registered medicine with a different indication (off-label), it is imperative that US-assisted contrast imaging is first tested for its feasibility and safety in a cohort of SS patients. Our study aimed to assess the safety, practical applicability, and initial efficacy of CEUSS in the salivary glands of patients with SS. The protocol of this study was published in 2020 [20].

## 2. Materials and Methods

### 2.1. Study Setting and Ethics

This prospective single-center trial was performed at the Amsterdam University Medical Centers, Departments of Oral and Maxillofacial Surgery/Oral Pathology, and Radiology and Nuclear Medicine, Location VU Medical Center Amsterdam, the Netherlands. The Medical Research Ethics Committee (MREC) of the Amsterdam UMC/VU Medical Center approved the study protocol (NL68283.029.19) and the study was conducted in accordance with the Declaration of Helsinki. Written informed consent was obtained from each patient. The study protocol was published in 2020 [20] and is registered in The Netherlands Trial Register (NTR7731).

### 2.2. Patient and Public Involvement

A patient advisory group from the Nationale Vereniging Sjögrenpatiënten (Dutch Association for Sjögren’s Patients) partnered with us for designing the study, preparing the informational material to support the intervention, and reviewing the burden of the intervention from the patient’s perspective.

### 2.3. Participants and Eligibility Criteria

Participants were recruited from our cohort of SS patients. To be eligible to participate in this study, a patient had to meet all of the following inclusion criteria: diagnosed with SS in agreement with the 2016 American College of Rheumatology–European League Against Rheumatism (ACR-EULAR) criteria [27]; age, ≥18 years and ≤75 years; a remaining unstimulated whole saliva flow (UWSF) rate of ≥0.02 mL/min and a remaining stimulated whole saliva flow (SWSF) rate of ≥0.10 mL/min. Patients were excluded when any of the following exclusion criteria applied: limited access to the salivary ductal orifice as determined during assessment, acute sialadenitis, severe illness or physical conditions interfering with the intervention, use of sialogogue medication (i.e., pilocarpine or cevimeline), history of head and neck cancer radiotherapy, or the presence of MALT lymphoma in the major salivary glands. Furthermore, patients suspected of acute coronary syndrome, recent percutaneous coronary intervention, acute or chronic severe (New York Heart Association (NYHA) class III/IV) heart failure, right-to-left shunts, severe pulmonary hypertension (pulmonary artery pressure > 90 mm Hg), uncontrolled hypertension, adult respiratory distress syndrome, or severe cardiac dysrhythmias were not eligible to participate [28,29,30,31]. The microbubbles used in this study were not used in combination with dobutamine in patients with cardiovascular instability, where dobutamine is contraindicated.

### 2.4. Intervention

CEUSS is a classic endoscopic technique combined with percutaneous US imaging specifically designed for application in the major salivary glands. This procedure is also described in our published protocol of this study [20]. After the orifice of the salivary gland duct to be treated was dilated using lacrimal probes (#0000 or #000 diameter), 0.5–1.0 mL of 4% (*w*/*v*) articaine with 1:100.000 adrenaline (Septanest, Septodont, Saint-Maru-des-Fosses, France) was injected submucosally near the papilla. Next, the ducts were dilated with probes with diameters increasing to #4. After dilatation, the endoscope was properly positioned in the salivary duct under visual guidance. Erlangen sialendoscopes with 0.8 and 1.1 mm outer diameters were used for this procedure (Karl Storz GmbH and Co., Tuttlingen, Germany). The intervention was started by flushing the salivary duct system and filling it with approximately 2 mL of saline to unfold the ducts. After the initial irrigation was performed, a mixture of 0.3 mL of a second-generation contrast agent (SonoVue^®^, Bracco, Milan, Italy) consisting of stabilized microbubbles of sulfur hexafluoride and 9.5 mL of 0.9% (*w*/*v*) NaCl was injected. During this procedure, continuous US imaging was performed to visualize the ducts and glands. Ultrasound imaging was performed on a General Electric Logiq E10 (GE Healthcare, Boston, MA, USA) using an ML6-15 linear probe at a fixed US resonance frequency of 15 MHz for diagnostic imaging and at 6 MHz and under low US mechanical index settings (e.g., MI:0.1) during irrigation with microbubbles to avoid disruption and premature activation of the microbubbles [26]. The distribution speed within the ductal system and glandular penetration of microbubbles during irrigation was visualized and images were continuously captured, allowing us to observe the efficacy of the sialendoscopic rinsing procedure on strictures (i.e., occlusions or blockades). During CEUSS, the contrast agent continuously drained in a retrograde manner from the ductal system via the ostium into the oral cavity and was removed from the oral cavity by suction. Therefore, the contrast agent was replenished regularly throughout the procedure. For this, a small volume of contrast agent was applied whenever the ducts collapsed. This strategy resulted in an average application rate of about 0.5 mL of irrigation fluid per minute [32]. Stronger and longer-lasting bursts of irrigation were necessary during endoscopy to flush out plaques and microsialoliths from the salivary duct system and to open strictures. On the surgeon’s instruction, the assisting nurse performed intermittent flushing by manual pressure on a 10 mL syringe.

### 2.5. Outcomes

As described in our published study protocol [20], the primary outcomes were the evaluation of safety and practical applicability of the experimental treatment. Safety was determined by unanticipated treatment-related mortality and the occurrence of adverse events (AEs) and serious adverse events (SAEs). AEs were defined as any undesirable experience occurring to a patient during the experimental treatment period and whether they were related to the investigational intervention. SAEs were defined as any untoward medical occurrence or effect that, at any dose, was either life-threatening (at the time of the event), required hospitalization, prolonged the existing in-patients’ hospitalization, resulted in persistent or significant disability or incapacity, or was a new event of the trial likely to affect the safety of the patients, such as an unexpected outcome of an adverse reaction. Practical applicability was defined as the successful completion of the experimental protocol during the procedure.

The secondary outcomes of CEUSS included measurements of unstimulated and stimulated whole saliva (UWS and SWS, respectively) flow rates and the clinical oral dryness score (CODS) [33,34]. Changes due to CEUSS in reported pain, mouthfeel, and clinical SS symptoms were determined by comparison to initial values using a set of validated questionnaires, i.e., the McGill pain questionnaire (MPQ) [35], the xerostomia inventory (XI) [36], and the European League Against Rheumatism (EULAR) Sjögren’s Syndrome Patient Reported Index (ESSPRI) [37,38]. The UWS and SWS flow rates, COD, MPQ, XI, and ESSPRI scores were recorded 4 weeks before CEUSS (T-4) and 1 (T1), 2 (T2), 8 (T8), and 16 (T16) weeks after CEUSS. Finally, salivary gland topographical alterations were evaluated by US using the Hočevar score [39]. The echostructure of the treated glands was graded at T-4, T1, and T16. Additionally, the secondary outcomes were described in our published protocol [20].

### 2.6. Participant Timeline

The study encompassed an enrolment and assessment period of 6 weeks and a patient follow-up period of 16 weeks. The schedule of enrolment, interventions, and assessments is presented in Table 1 [20].

### 2.7. Data Collection Methods

The data collection methods are previously described in our publication of the study protocol [20].

#### 2.7.1. Saliva Flow Rates and Analysis

Each patient was instructed to refrain from drinking, eating, chewing, brushing their teeth, and smoking for 90 min before each visit. To minimize diurnal variation, all appointments for each patient were at the same time of day and in the same room (temperature 21 ± 2 °C, humidity 50–60%). UWS and SWS samples were collected every 30 s over 5 min periods in separate pre-weighed containers. For the UWS samples, each patient was instructed to start collecting saliva immediately after an initial swallow and expectoration. For the SWS samples, patients were instructed to chew a 5 × 5 cm sheet of paraffin (ParafilmM, Pechiney, Chicago, IL, USA) and then expectorate every 30 s for 5 min. The patients were instructed to chew at a rate of 60 strokes/min, indicated by a metronome, to reflect a normal chewing rate. Each container was reweighed after saliva collection and the weight of the empty container was subtracted to determine UWS and SWS flow rates (mL/min; assuming 1 g = 1 mL) [40]. The same observer (AM) performed all assessments.

#### 2.7.2. Xerostomia Inventory (XI) Score

The summated XI score is based on an 11-item internationally validated questionnaire about oral dryness and mouthfeel. A five-point Likert scale is used to indicate symptom frequency. The values from the questions are summed to give a total XI score of 11 (no dry mouth) to 55 (extreme dry mouth) [36].

#### 2.7.3. European League against Rheumatism (EULAR) Sjögren’s Syndrome Patient Reported Index (ESSPRI)

The disease symptoms (pain, fatigue, and dryness) were assessed using the 10-point scale ESSPRI patient-administered questionnaire. The ESSPRI has a high sensitivity for the detection of changes in symptoms after a therapeutic intervention is performed. Only the dryness subscale was included in the analysis. A change of two or more points is considered clinically relevant [37].

#### 2.7.4. Clinical Oral Dryness Score (CODS)

The CODS is a validated clinical guide designed to assess oral dryness using clinical and visual inspection of the oral cavity. It includes 10 clinical signs of oral dryness, such as the presence of frothy saliva and stickiness of the dental mirror to the tongue and buccal fold. The values for each of the characteristics are summed to result in a score ranging from 0 (no oral dryness) to 10 (extreme oral dryness) [33,34].

#### 2.7.5. Pain Score

The Dutch version of the McGill pain questionnaire (MPQ) is a three-part pain assessment tool that measures several dimensions of the patient’s pain experience. The first part consists of an anatomic drawing of the human form on which the patient marks where their pain is located. The second part of the MPQ is a visual analogue scale (VAS) that allows the patient to record the intensity level of their current pain experience. The third part of the MPQ is a verbal pain descriptor inventory consisting of 72 descriptive adjectives. The patient is asked to review this list of pain descriptors and circle the ones that serve to best describe their current pain experience. Each part or dimension of the MPQ is individually scored and recorded, along with a cumulative total score [35].

#### 2.7.6. Evaluation of the Major Salivary Glands by US

At T-4, T1, and T16 the salivary glands were examined using salivary gland ultrasound (SGUS). The Hočevar scoring system was used to investigate: (1) parenchymal echogenicity compared with the thyroid gland, graded 0–1; (2) homogeneity, graded 0–3; (3) the presence of hypo-echogenic areas, graded 0–3; (4) hyper-echogenic reflections, graded 0–3 in parotid glands and 0–1 in the submandibular glands; and (5) clearness of the salivary gland border, graded 0–3, in both the parotid and submandibular salivary glands. The total US score is the sum of these five domains and can range from 0 to 48 [39]. An experienced ultrasonographer (JB), who was blinded to the patients’ data, performed all US examinations.

### 2.8. Sample Size Analysis and Statistical Methods

Usually, 10–20 patients are investigated in a phase I trial to confirm the occurrence of toxic effects or (S)AEs that are anticipated to be <20%. A descriptive analysis of the primary outcome measures (AEs and SAEs) was performed. The mean and standard deviation are reported for data with a normal distribution. The median and interquartile ranges (IQR) are reported for data with a non-normal distribution. Additionally, the mean and standard deviation are reported for non-normally distributed data to compare relatively small differences and to enable comparisons with the existing literature.

Wilcoxon signed-rank tests (for data without normal distribution) or analysis of variance for repeated measurements followed by a Bonferroni post hoc test (for data with a normal distribution; assumption of sphericity was assessed with Mauchly’s test) were used to examine the differences between baseline and subsequent time points for the secondary outcome measures. When sphericity was violated, a Greenhouse–Geisser correction was used. The data were analyzed with SPSS 27.0 (IBM, Armonk, NY, USA). A *p* value of 0.05 or lower was considered statistically significant.

## 3. Results

Ten patients of the eleven included patients completed the study between September 2020 and October 2021 (last follow-up). During follow-up, between weeks 1 and 8, one patient left the study because she no longer wanted to participate but did not give any reason for her withdrawal (Figure 1).

The characteristics of the study population and the baseline values for all parameters are presented in Table 2. All 10 patients were female, since SS predominantly affects women, and had a mean age of 47.7 years (SD: 18.7; range: 20–78 years). Furthermore, most patients had secondary SS (6 of 10). The median disease duration (since diagnosis) was 4 years. The analysis of the data for normality revealed that XI, ESSPRI, COD, and pain scores were normally distributed and UWS and SWS flow rates and Hočevar scores were not normally distributed (Shapiro–Wilk; *p* < 0.05). The UWS and SWS flow rates are presented in Table 3 and the XI, ESSPRI, COD, and pain scores are presented in Table 4. In Table 5, the total Hočevar score and the scores of its subdomains are presented. Figure 2A–D and the Appendix A illustrate the variable penetration patterns of microbubbles into glandular tissue, as observed after the irrigation procedure.

### 3.1. Primary Outcome Measures

The procedure was technically feasible in all 10 patients. The mean operating time was 33 min (range 21–50 min). Furthermore, no SAEs nor systemic reactions related to the procedure were reported during the study period. The main AEs were postoperative pain (2 patients) and swelling of the glands (2 patients), which lasted for a maximum of 1 week. Furthermore, one patient experienced a tingling sensation in salivary passages when eating during the first week following the procedure. Medication (acetaminophen) was sufficient to treat pain, and post-procedural swelling disappeared without treatment.

### 3.2. Secondary Outcome Measures

The secondary outcomes for all 10 patients are presented in Table 3 and Table 4. Pain scores were measured on a visual analog scale and the mean pain score was higher at T1, T2, and T8 but lower at T16 compared to baseline (T-4). These differences were not statistically significant (F(4,36) = 0.539, *p* = 0.71). The MPQ also determined the location(s) where the patients experienced pain. Before CEUSS, two patients reported pain originating from the parotid and submandibular glands. After CEUSS, at T1, this pain was no longer reported. In contrast, two other patients reported pain in the parotid glands 1 week after CEUSS and this pain was accompanied by swelling of the affected glands. At T2, this pain was no longer reported.

At T16 compared to T-4, median UWS and SWS flow rates increased from 0.10 to 0.24 mL/min and from 0.41 to 0.66 mL/min, respectively. A significantly higher UWS flow rate compared to baseline (Median: 0.10 mL/min) was found at T1 (Median: 0.18 mL/min; *p* = 0.037), T2 (Median: 0.22 mL/min; *p* = 0.007), and T8 (Median: 0.22 mL/min; *p* = 0.028). For the SWS flow rate, a significantly higher improvement compared to T-4 (Median: 0.41 mL/min) was only found at T8 (Median: 0.61 mL/min; *p* = 0.047). The percentage of patients in whom any improvement in saliva secretion rate was observed after 16 weeks was 60% for UWS and 70% for SWS flow rates. The percentage of patients who regained an adequate salivary flow rate (defined for UWS > 0.1 mL/min and SWS > 0.5 mL/min) after 16 weeks was 60% for UWS and 70% for SWS flow rates.

XI, ESSPRI, and COD scores showed a significant improvement at most follow-up visits (Table 4). The mean XI scores were significantly lower after intervention at all time points compared with baseline, suggesting that CEUSS resulted in a reduced dry mouth feeling [F(4,36) = 11.85, *p* < 0.001]. Post hoc tests revealed that the mean XI score at T16 was 11 (*p* = 0.021; 95% CI:1.49–20.51) points lower compared to T-4. Please note that a minimally important difference for the alleviation of symptoms is not yet determined for the XI scores [41].

The dryness domain of the ESSPRI was lower compared with the baseline from T2 onwards and the mean dryness domain of ESSPRI scores differed significantly between these time points and T-4 [F(4,36) = 7.581, *p* < 0.001]. Post hoc tests showed the mean dryness domain ESSPRI score at T2 to be 3.00 (95% CI: 0.54–5.46, *p* = 0.015) points lower than at T-4. At 16 weeks, the mean dryness domain ESSPRI score was 4.10 (95% CI: 1.74–6.46, *p* < 0.001) points lower than at T-4. In addition, the mean dryness domain ESSPRI score at T8 was found to differ significantly from T-4. A change of two or more points is considered clinically relevant.

The mean CODS decreased after intervention and was found to differ significantly between time points T2-T16 and T-4 [F(4,13.54) = 14.245, *p* < 0.001]. Because there was a violation of the assumption of sphericity (χ^2^(9) = 22.36, *p* = 0.01), a Greenhouse–Geisser correction was applied. Post hoc tests revealed that the CODS decreased by an average of 1.67 (95% CI: 0.39, 2.94, *p* = 0.011) points between T2 and T-4. The mean CODS at T16 was 2.22 (95% CI: 0.13, 4.32, *p* = 0.036) points lower than T-4. In addition, a significant decrease compared with baseline was also found at T8.

The median total Hočevar score did not change significantly (*p* = 0.20). When looking at the different subdomains, the Hočevar score also showed no significant improvement in any of the subdomains (Table 5).

## 4. Discussion

The results of our study indicate that CEUSS is a safe and technically feasible procedure in SS patients. No (serious) adverse events from the contrast media were observed. Furthermore, the procedure was easily implemented and did not cause additional pain or inconvenience for the patients compared to classic sialendoscopy [42]. Moreover, our results correspond with a previous study using the same non-ionic contrast agent in salivary glands of patients without SS [43]. Furthermore, the CEUSS procedure is inexpensive, not time-consuming (mean: 33 min; range 21–50 min), and might be performed multiple times under local anesthesia [43,44].

The results suggest that CEUSS reduced both hyposalivation and xerostomia, at least comparably to the effect of classic sialendoscopy [15]. However, contrast-enhanced sialendoscopy may have an advantage over classic sialendoscopy because of the dynamic real-time information obtained during the procedure. This information provides the surgeon with peri-operative feedback about the penetration of microbubbles and successful recanalization of the ductal system and the salivary parenchyma by the rinsing solution. Furthermore, it is possible to observe the dissolution of ductal obstructions such as strictures [26]. When we compare the results of our current study with our previous study investigating the effect on saliva secretion of classic sialendoscopy using saline as the irrigation solution [14], it was found that the median UWS flow rate increased from 0.10 mL/min to 0.13 mL/min in the latter study, while in the current study, the median UWS flow rate increased from 0.10 mL/min to 0.24 mL/min. Similarly, the SWS flow rate increased from 0.25 mL/min to 0.33 mL/min and from 0.41 mL/min to 0.66 mL/min after classic sialendoscopy and CEUSS, respectively. It should be noted that these differences are not statistically significant and that the baseline SWS flow rate of the participants in the current study was significantly higher than in our previous study. This makes comparison of the studies difficult.

The increase in salivary secretion can be explained by dilatation prior to, and during, the endoscopic procedure, as this may open ductal strictures and remove debris such as microsialoliths and mucus plugs [45]. Additionally, other mechanisms that may explain any beneficial effect of ductal irrigation that can also apply to sialendoscopic treatments were suggested by Aframian et al. For example, stress conditioning could be induced by dilatation. It is suggested that exposure of salivary glands to stress results in the propagation of salivary gland stem cell capabilities due to cellular plasticity in the glands’ parenchyma. This could promote salivary gland repair [16,17]. Future (animal) studies should explore the effect of sialendoscopy on the ductal and acinar compartments to reveal the mechanism that accounts for this finding.

Furthermore, CEUSS has potential value in the diagnostic work-up of SS and could possibly contribute to the assessment of disease severity and progression, since an assessment of the peripheral ductal system and parenchyma is possible in contrast to classic sialendoscopy. Classic sialendoscopy provides the possibility to inspect the ductal system with direct vision but does not play a role in diagnosing SS or assessing disease severity. In contrast, evidence is emerging that salivary gland ultrasound (SGUS) can effectively assess SS severity and may contribute to diagnosing SS [46,47]. The combination of the presence of hypo-echogenic areas in SGUS images and anti-SSA antibodies was shown to be highly predictive for classifying a patient with pSS. [48]. Unfortunately, ultrasound in general is considered to be an operator-dependent diagnostic method. Moreover, with SGUS, the salivary duct is only visible in the case of obstruction, when the ductal system is in front of a stone or the ductal stricture is dilated. As a result, concerns were raised regarding the reliability of SGUS [48]. CEUSS could perhaps increase the specificity of SGUS by providing information on patency from even the smallest ducts outside the scope of obstruction. With CEUSS, it is possible to clearly visualize both the ductal system and the gland by the distribution of microbubbles (Figure 2). Before CEUSS can be used for this, we should know the distribution of microbubbles in healthy salivary glands, and a scoring system should be developed to quantify these patterns of distribution.

The Hočevar scoring system [49] was applied to detect topographic changes of the salivary glands after CEUSS compared to baseline. For this, parenchymal echogenicity, homogeneity, the presence of hypo-echogenic areas, hyper-echogenic reflections, and the clarity of the salivary gland border in both the parotid and submandibular salivary glands were scored. A total Hočevar score of ≥15 is considered positive for SS [50]. In our study, three participants did not fulfill this criterion at T-4 (range: 4–6) despite a positive SS diagnosis. The seven other participants had a total Hočevar score between 18 and 37. In previous studies, it was suggested that salivary gland US grading using the Hočevar score may help predict outcomes of treatment for impaired salivary function in patients with SS [51]. Recovery of the salivary gland parenchyma may indicate the success of SS treatments, along with enhanced salivary secretion and the reversal of xerostomia. Consequently, it is possible and feasible to repeat CEUSS, potentially quite frequently, to observe the persistent patency of the ducts and the state of recovery of the glandular tissue. Unfortunately, we found that using Hočevar scores to predict the ability of CEUSS to reverse the SS phenotype in patients did not correlate well with an increase in salivary function and a reduction in xerostomia. The development of a microbubble perfusion score for salivary glands may overcome this contradiction. Thus, a comparison between microbubble perfusion, Hočevar score, and salivary function may have greater prognostic value for salivary gland function after CEUSS.

In the current study, microbubbles served two purposes, as described in our previously published study protocol [20]: (1) the μm size of microbubbles facilitates penetration into virtually all ducts, thereby allowing non-invasive high-resolution US imaging of the ductal system and obstructions (where microbubbles accumulate), and (2) the effect of CEUSS can be imaged by the re-infusion of microbubbles to assess the ‘new’ distribution pattern. Thus, if ductal obstructions are not sufficiently resolved, re-treatment can be performed in the same session. When activating microbubbles locally (i.e., when the microbubbles burst) during CEUSS, by using harmonic US waves focally, obstructions within the ductal system may be resolved even more effectively. This could be a subject of future animal and human studies as the activation of microbubbles is not yet tested for safety in salivary glands.

In future studies, CEUSS could also be used to visualize, observe, and confirm the distribution of drugs within the salivary glands. This could be implemented by developing drug-loaded microbubbles for image-guided and ultrasound-triggered drug delivery [52]. Ultrasound-triggered, microbubble-assisted drug delivery is currently being investigated for several applications [53,54,55]. Another approach could be the application of CEUSS before local drug administration. Because CEUSS will remove any obstructions in the ductal system, CEUSS could facilitate optimal penetration of therapeutic compounds into the ductal system of salivary glands and other organs. We expect that microbubble concentrations in salivary glands, visualized by ultrasound, will correspond with their therapeutic compound levels.

## 5. Conclusions

From this study, it can be concluded that CEUSS is safe and feasible in SS patients. It could have the potential to increase salivary secretion and reduce xerostomia, but this needs further investigation in our next study, in which CEUSS will be compared with a placebo group using a larger sample size. Ultimately, we would like to have this application of Sonovue microbubbles classified as on-label.

## Figures and Tables

**Figure 1 jcm-12-04152-f001:**
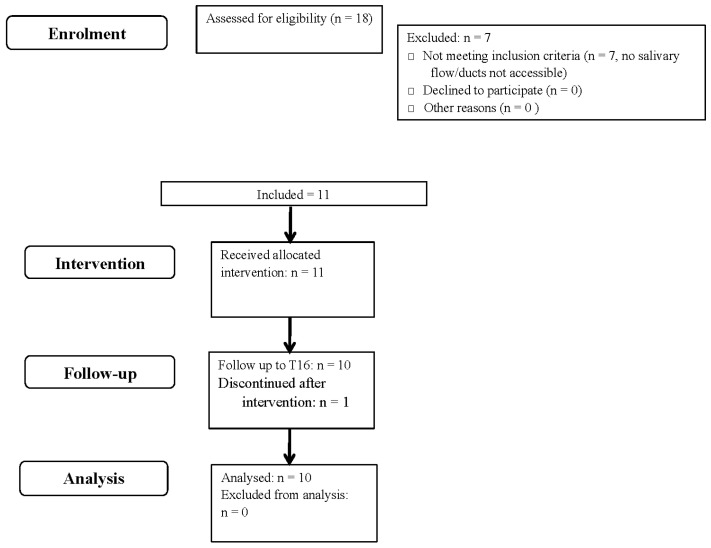
Flow diagram of the allocation of participants.

**Figure 2 jcm-12-04152-f002:**
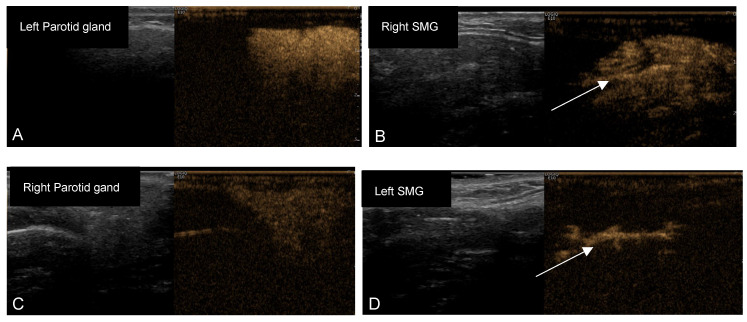
Images of (**A**) left parotid gland with complete homogeneous dense penetration of microbubbles. (**B**) Right submandibular gland (SMG) with complete, inhomogeneous dense penetration of microbubbles. (**C**) Right parotid gland with complete homogenous penetration of microbubbles with moderate density. (**D**) Left SMG with no penetration of microbubbles. Only the main duct is visible. Yellow areas: microbubbles; Arrow: Wharton’s duct. Please note: the images are from four different patients.

**Table 1 jcm-12-04152-t001:** Schedule of enrolment, interventions, and assessments.

	Enrolment	Baseline Assessment	Intervention Post-Intervention Assessment				
**Timepoint**	**T-6**	**T-4**	**T0**	**T1**	**T2**	**T8**	**T16 (Closeout)**
**Enrolment:**							
Eligibility screen	X						
Informed consent		X					
**Intervention:**							
CEUSS			X				
**Assessments:**							
Safety				X	X	X	X
Technical feasibility and applicability			X				
UWSF, SWSF	X	X		X	X	X	X
CODS		X		X	X	X	X
XI		X		X	X	X	X
ESSPRI		X		X	X	X	X
MPQ		X		X	X	X	X
Salivary gland topography with SGUS		X		X			X

Abbreviations: CEUSS: contrast-enhanced ultrasound sialendoscopy, UWSF: unstimulated whole saliva flow, SWSF: stimulated whole saliva flow, CODS: clinical oral dryness score, XI: xerostomia inventory, ESSPRI: EULAR Sjögren’s Syndrome Patient Reported Index, MPQ: McGill pain questionnaire, SGUS: salivary gland ultrasound. T-6: 6 weeks before intervention; T-4: 4 weeks before intervention; T0: intervention; T1, T2, T8, T16: 1, 2, 8, and 16 weeks after CEUSS, respectively.

**Table 2 jcm-12-04152-t002:** Characteristics of the study population (*n* = 10) and baseline values for all parameters. Baseline measurements are also subdivided according to the diagnosis of primary SS (*n* = 4) or secondary SS (*n* = 6).

		Mean (S.D.)	Median (IQR)
**Patient variables**	Age, years	47.7 (18.7)	49.5 (28.5–62.5)
	Female gender, *n*	10	
**Disease variables**	Disease duration, years ^a^	8.3 (8.4)	4.0 (2.0–17.5)
	Primary SS, *n* ^b^	4	
	Secondary SS, *n* ^b^	6	
	Positive salivary gland biopsy, *n*	4	
	Autoantibodies to anti-SSA, *n*	7	
**Baseline measurements**	UWSF (mL/min)	0.13 (0.09)	0.10 (0.07–0.22)
	pSS	0.12 (0.06)	0.09 (0.15–0.50)
	sSS	0.15 (0.11)	0.16 (0.03–0.25)
	SWSF (mL/min)	0.50 (0.43)	0.41 (0.18–0.75)
	pSS	0.39 (0.27)	0.32 (0.17–0.67)
	sSS	0.59 (0.52)	0.49 (0.15–0.97)
	XI Score	45.20 (3.55)	
	pSS	45.75 (3.86)	
	sSS	44.83 (3.66)	
	ESSPRI (all domains) ^c^	21.5 (5.3)	
	pSS	17.8 (5.1)	
	sSS	21.5 (5.3)	
	ESSPRI (dryness domain)	7.8 (1.6)	
	pSS	7.5 (1.7)	
	sSS	8.0 (1.7)	
	CODS	4.0 (1.6)	
	pSS	4.5 (1.3)	
	sSS	3.8 (1.7)	
	Hoçevar score	20.4 (11.9)	21.5 (7.0–29.5)
	pSS	21.8 (13.0)	24.5 (8.3–32.5)
	sSS	19.5 (13.3)	20.0 (7.0–30.3)
	Pain score	38.4 (32.8)	
	pSS	9.3 (12.3)	
	sSS	57.8 (26.7)	

Mean (S.D.) and median (interquartile range; IQR) values are presented for data with a non-normal distribution. ^a^ Disease duration is years since diagnosis. ^b^ Classified according to the 2016 American European Consensus Group Criteria (AECG); 5 patients classified as secondary SS had rheumatoid arthritis, and 1 patient had systemic lupus erythematosus (SLE). ^c^ Defined as the total ESSPRI score divided by 3. Abbreviations: S.D.: standard deviation, IQR: interquartile range, SS: Sjögren’s Syndrome, UWSF: unstimulated whole saliva flow, SWSF: stimulated whole saliva flow, XI: xerostomia inventory, CODS: clinical oral dryness score, pSS: primary Sjögren’s Syndrome, sSS: secondary Sjögren’s Syndrome.

**Table 3 jcm-12-04152-t003:** UWS and SWS flow rates at baseline and subsequent time points for all patients (*n* = 10), and subdivided according to the diagnosis of primary SS (*n =* 4) or secondary SS (*n =* 6).

		Median (IQR)	Mean (S.D.)	*p* Value (Compared to Baseline)
**UWSF (mL/min)** **All patients** ** *n* ** ** = 10**	**T-4**	0.10 (0.07–0.22)	0.13 (0.09)	--
	**T1**	0.18 (0.06–0.35)	0.26 (0.29)	0.037 *
	**T2**	0.22 (0.10–0.49)	0.29 (0.24)	0.007 *
	**T8**	0.22 (0.15–0.50)	0.38 (0.44)	0.028 *
	**T16**	0.24 (0.09–0.40)	0.26 (0.21)	0.074
**UWSF (mL/min)** **Primary SS** ** *n* ** ** = 4**	**T-4**	0.09 (0.08–0.17)	0.12 (0.06)	--
	**T1**	0.10 (0.05–0.27)	0.14 (0.13)	0.72
	**T2**	0.18 (0.12–0.37)	0.22 (0.14)	0.07
	**T8**	0.20 (0.08–0.37)	0.22 (0.16)	0.14
	**T16**	0.18 (0.08–0.34)	0.20 (0.14)	0.47
**UWSF (mL/min)** **Secondary SS** ** *n* ** ** = 6**	**T-4**	0.16 (0.03–0.25)	0.15 (0.11)	--
	**T1**	0.28 (0.06–0.57)	0.34 (0.34)	0.028 *
	**T2**	0.25 (0.08–0.69)	0.33 (0.30)	0.046 *
	**T8**	0.27 (0.15–0.91)	0.49 (0.54)	0.12
	**T16**	0.26 (0.10–0.54)	0.30 (0.25)	0.12
**SWSF (mL/min)** **All patients** ** *n* ** ** = 10**	**T-4**	0.41 (0.18–0.75)	0.50 (0.43)	--
	**T1**	0.60 (0.20–1.15)	0.70 (0.57)	0.06
	**T2**	0.55 (0.24–0.93)	0.68 (0.55)	0.07
	**T8**	0.61 (0.27–0.97)	0.73 (0.64)	0.047 *
	**T16**	0.66 (0.17–0.84)	0.66 (0.52)	0.33
**SWSF (mL/min)** **Primary SS** ** *n* ** ** = 4**	**T-4**	0.32 (0.17–0.67)	0.39 (0.27)	--
	**T1**	0.26 (0.16–0.61)	0.34 (0.26)	0.47
	**T2**	0.36 (0.16–0.46)	0.33 (0.16)	0.72
	**T8**	0.35 (0.19–0.59)	0.38 (0.22)	0.72
	**T16**	0.35 (0.15–0.66)	0.40 (0.27)	1.00
**SWSF (mL/min)** **Secondary SS** ** *n* ** ** = 6**	**T-4**	0.49 (0.15–0.97)	0.59 (0.52)	--
	**T1**	0.83 (0.44–1.61)	0.93 (0.62)	0.028 *
	**T2**	0.78 (0.51–1.50)	0.92 (0.61)	0.028 *
	**T8**	0.68 (0.48–1.76)	0.97 (0.73)	0.028 *
	**T16**	0.70 (0.49–1.37)	0.85 (0.59)	0.25

The data are expressed as medians (interquartile range; IQR) and mean (standard deviation; S.D.). UWSF: unstimulated whole saliva flow. SWSF: chewing-stimulated whole saliva flow. SS: Sjögren’s Syndrome T-4: 4 weeks before intervention; T1, T2, T8, T16: 1, 2, 8 and 16 weeks after sialendoscopy, respectively. * Significant.

**Table 4 jcm-12-04152-t004:** Xerostomia inventory, ESSPRI, clinical oral dryness and pain scores at baseline and subsequent time points.

		Mean (S.D.)	*p* Value (Compared to Baseline)	95% Ci	
				Upper Limit	Lower Limit
**Xerostomia Inventory**	**T-4**	45.20 (3.55)	--	--	--
	**T1**	37.00 (8.33)	0.04 *	0.31	16.10
	**T2**	37.80 (7.57)	0.031 *	0.57	14.23
	**T8**	34.40 (7.027)	0.003 *	3.89	17.71
	**T16**	34.20 (8.84)	0.021 *	1.49	20.51
**ESSPRI (all domains)**	**T-4**	21.50 (5.26)	--	--	--
	**T1**	16.30 (4.22)	0.018 *	0.81	9.59
	**T2**	17.10 (5.69)	0.001 *	1.93	6.87
	**T8**	16.80 (4.76)	0.044 *	0.10	9.30
	**T16**	16.80 (5.63)	0.10	−0.66	10.06
**ESSPRI (dryness domain)**	**T-4**	7.80 (1.62)	--	--	--
	**T1**	5.80 (2.44)	0.34	−0.96	4.96
	**T2**	4.80 (2.35)	0.015 *	0.54	5.46
	**T8**	4.50 (2.01)	0.015 *	0.60	6.00
	**T16**	3.70 (1.83)	0.001 *	1.74	6.46
**Clinical Oral Dryness**	**T-4**	4.00 (1.58)	--	--	--
	**T1**	3.33 (1.23)	0.22	−0.24	1.57
	**T2**	2.33 (1.00)	0.011 *	0.39	2.94
	**T8**	1.89 (0.60)	0.017 *	0.37	3.85
	**T16**	1.78 (0.67)	0.036 *	0.13	4.32
**VAS pain score**	**T-4**	38.4 (32.8)	--	--	--
	**T1**	40.8 (32.0)	1.00	−41.55	36.75
	**T2**	48.0 (34.3)	1.00	−48.02	28.82
	**T8**	44.2 (35.1)	1.00	−46.03	34.43
	**T16**	34.5 (31.7)	1.00	−17.77	25.57

The data are expressed as means (S.D.) and 95% confidence interval (CI) for XI, ESSPRI (total score and dryness domain only), COD and VAS pain scores for all time points. The Bonferroni correction was applied to *p* values. * Significant. XI: xerostomia inventory. CODS: clinical oral dryness score. ESSPRI: European Alliance of Associations for Rheumatology Sjögren’s Syndrome Patient Reported Index. VAS: visual analog scale. T-4: 4 weeks before intervention; T1, T2, T8, T16: 1, 2, 8, and 16 weeks after CEUSS, respectively.

**Table 5 jcm-12-04152-t005:** Hoçevar median and mean total scores (sum of four glands, five components) and median and mean scores for each component.

		Median (IQR)	Mean (S.D.)	*p* Value (Compared to Baseline)
**Total Hoçevar score**	**T-4**	21.5 (7.0–29.5)	20.4 (11.9)	--
	**T1**	22.0 (6.0–27.5)	19.1 (11.4)	0.29
	**T16**	20.0 (6.0–29.0)	19.3 (11.8)	0.20
**Parenchymal echogenicity**	**T-4**	0.5 (0.0–2.3)	1.2 (1.5)	--
	**T1**	0.5 (0.0–2.3)	1.1 (1.5)	0.66
	**T16**	0.5 (0.0–2.3)	1.2 (1.6)	1.0
**Homogeneity**	**T-4**	5.5 (2.3–10.0)	6.1 (3.5)	--
	**T1**	6.0 (2.9–8.75)	6.0 (3.5)	0.71
	**T16**	5.5 (2.0–8.8)	6.0 (3.7)	0.71
**Presence of hypo-echogenic areas**	**T-4**	8.0 (2.3–12.0)	7.6 (4.5)	--
	**T1**	8.0 (2.0–11.0)	7.3 (4.2)	0.32
	**T16**	7.5 (2.0–11.0)	7.3 (4.1)	0.32
**Hyper-echogenic reflections**	**T-4**	4.0 (0.0–5.0)	3.1 (2.3)	--
	**T1**	2.5 (0.0–4.3)	2.3 (2.2)	0.11
	**T16**	2.0 (0.0–4.3)	2.3(2.5)	0.11
**Clearness of the salivary gland border**	**T-4**	2.0 (0.0–4.3)	2.4 (2.1)	--
	**T1**	2.0 (0.0–4.3)	2.4 (2.2)	1.00
	**T16**	2.0 (0.0–4.5)	2.5 (2.3)	0.32

The data are expressed as medians (interquartile range; IQR) and means (standard deviation; S.D.). T-4: 4 weeks before sialendoscopy; T1, T16: 1 and 16 weeks after sialendoscopy, respectively.

## Data Availability

The data that support the findings of this study will be openly available in the Dryad Digital Repository.

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
