# Peer review of "Intraoperative Visualization and Treatment of Salivary Gland Dysfunction in Sjögren’s Syndrome Patients Using Contrast-Enhanced Ultrasound Sialendoscopy (CEUSS)"

_jcm, 2023, doi:10.3390/jcm12124152_

Round 1

Reviewer 1 Report

The work by Karagozoglu et is a well-written and interesting about the use of Contrast-En-3 hanced Ultrasound Sialendoscopy (CEUSS) in the treatment of patient with Sjogren Syndrome. The article is catchy, but in my opinion it seems that could be improved:

The Introduction section is well-written but is missing of a background about CEUSS and about the scientific rationale for the use of this tool in this field of research.

The M & M are well-edited with a high scientific value and don't need further improvement. 

Table 1. Seems a bit confusing. I would try to make the table more clear and easy to read

Conclusion section needs to be improved describing what path should take further studies to validate this technique, what could possibly be improved and what seem the limitation to the use of this tool. 

Author Response

Reviewer 1.

The work by Karagozoglu et is a well-written and interesting about the use of Contrast-Enhanced Ultrasound Sialendoscopy (CEUSS) in the treatment of patient with Sjogren Syndrome. The article is catchy, but in my opinion, it seems that could be improved:

Question: The Introduction section is well-written but is missing of a background about CEUSS and about the scientific rationale for the use of this tool in this field of research.

Answer: We believe that the background of CEUSS and its scientific rationale are well covered in the Introduction. In lines 73-85 we describe the limitations of the predecessor of CEUSS and subsequently we discuss in lines 86-96 how CEUSS can overcome these limitations and what the clinical applicability of CEUSS is. To emphasize these points better, we have adjusted the text in these lines.

The M & M are well-edited with a high scientific value and don't need further improvement. 

Question: Table 1. Seems a bit confusing. I would try to make the table clearer and easier to read

Answer:  We have modified the content of the table, trying to make it clearer. The overall design of the table has been maintained because this is the design recommended in the SPIRIT guideline. The introduction of T-4 for the baseline measurement 4 weeks before the intervention, and T0 for the intervention itself in this Table, have also been used throughout the manuscript to indicate these time points.

Question: Conclusion section needs to be improved describing what path should take further studies to validate this technique, what could possibly be improved and what seem the limitation to the use of this tool. 

Answer: We have modified the conclusion as suggested by the reviewer (lines 487 – 489). What could be improved and what the limitations are of this technique are already addressed in the Discussion section of our manuscript.

Reviewer 2 Report

The study is about intraoperative visualization and treatment of salivary gland dysfunction in Sjoegren’s syndrome patients using Contrast-enhanced Ultrasound Sialendoscopy (CEUSS) to test its safety and feasibility. CEUSS was found to be technically feasible in all patients, and there were no serious adverse events or systemic reactions related to the procedure. The main adverse events observed were postoperative pain and swelling in some patients. After 8 weeks of CEUSS, both UWS and SWS flow rates had increased significantly, and after 16 weeks, the XI was reduced. Therefore, CEUSS was considered a safe and feasible treatment for SS patients and has the potential to improve salivary secretion and reduce xerostomia, which requires further investigation.

But there are some considerations:

Line 76: Some study methods are presented in the introduction.

Line 120: Why was a control group not included with classical sialendoscopy with the same inclusion and exclusion criteria as the study patients?

Line 398: Is the comparison between the previous study's results on saliva secretion of classic sialendoscopy using saline as the irrigation solution, where the median UWS flow rate increased from 0.10 ml/min to 0.13 ml/min, and the current study, where the median UWS flow rate increased from 0.10 ml/min to 0.24 ml/min, statistically significant?

Line 403: In the previous study's classical sialendoscopy, the SWS flow rate increased from 0.25 ml/min to 0.33 ml/min, whereas in the presented study, it increased from 0.41 ml/min to 0.66 ml/min. Can these groups be compared? The groups are different because the initial value was much worse in the classical sialendoscopy study group.

Line 455: The results seem promising, but the study group needs to be larger, and there is a need for control groups (classic sialendoscopy) using saline solution and also a control group with the instillation of steroids.

LIne 416-418:

"Furthermore, CEUSS has potential value in the diagnostic work-up of SS and could contribute to the assessment of disease severity and progression, since an assessment of the peripheral ductal system and parenchyma is possible in contrast to classic sialendoscopy." - this statement is nco tsupported by the findings. Did the authors mean the central duct system? Moreover in just 10 cases one cannot assess the severity an progression o fandisease. Do we know at all how the contrast agents works in normal glands and how the "normal" distribution is?

Diagnosing an staging of lymphom is also not possible with CEUSS.

It is not shown that CEUSS is less operator dependent than Ultrasoudn alone

Author Response

Reviewer 2.

The study is about intraoperative visualization and treatment of salivary gland dysfunction in Sjoegren’s syndrome patients using Contrast-enhanced Ultrasound Sialendoscopy (CEUSS) to test its safety and feasibility. CEUSS was found to be technically feasible in all patients, and there were no serious adverse events or systemic reactions related to the procedure. The main adverse events observed were postoperative pain and swelling in some patients. After 8 weeks of CEUSS, both UWS and SWS flow rates had increased significantly, and after 16 weeks, the XI was reduced. Therefore, CEUSS was considered a safe and feasible treatment for SS patients and has the potential to improve salivary secretion and reduce xerostomia, which requires further investigation.

But there are some considerations:

Question: Line 76: Some study methods are presented in the introduction.

Answer: In the Introduction, we describe classic sialendoscopy. We think that this information is needed to provide readers who are not experts on this topic with a background of the limitations of classic sialendoscopy. Therefore, we like to maintain this information in the Introduction

Question: Line 120: Why was a control group not included with classical sialendoscopy with the same inclusion and exclusion criteria as the study patients?

Answer: we would potentially have been possible to have added a control group with classical sialendoscopy to this study. However, due to the off-label use (see lines 98 – 102) of the microbubbles and the application in a specific patient category, we were forced to first collect data on safety and feasibility. This information is usually collected using a study design as applied by us. We are now planning a study in which comparisons will be made with other interventions and a control group using a larger sample size (see lines 487 – 488). In this way we hope to further validate this treatment and ultimately to add the salivary glands as an approved indication for Sonovue microbubbels.

Question: Line 398: Is the comparison between the previous study's results on saliva secretion of classic sialendoscopy using saline as the irrigation solution, where the median UWS flow rate increased from 0.10 ml/min to 0.13 ml/min, and the current study, where the median UWS flow rate increased from 0.10 ml/min to 0.24 ml/min, statistically significant?

Answer: The difference in UWSF between the current study and our previous study does not appear to be significant. We have added text about this in lines 410-413

Question: Line 403: In the previous study's classical sialendoscopy, the SWS flow rate increased from 0.25 ml/min to 0.33 ml/min, whereas in the presented study, it increased from 0.41 ml/min to 0.66 ml/min. Can these groups be compared? The groups are different because the initial value was much worse in the classical sialendoscopy study group.

Answer: We agree with the reviewer that these 2 groups are difficult to compare due to the large difference in baseline values. These values ​​also appear to differ significantly from each other. Possibly, patients with a higher baseline value would show a greater response to our treatment. It should be noted that we believe that the UWSF is more relevant for the prevention of xerostomia than SWSF. This is due to the composition of the UWSF saliva. We have addressed this in lines 410- 413.

Question: Line 455: The results seem promising, but the study group needs to be larger, and there is a need for control groups (classic sialendoscopy) using saline solution and also a control group with the instillation of steroids.

Answer: We agree with the reviewer that a comparison should be made with a control group in order to be able to make more pronouncements on this. Unfortunately, as described above, regulations forced us to first investigate the safety and applicability of microbubbles in salivary glands of SS patients, as we used these bubbles off-label. This information is usually collected using a study design as applied by us.

Question: Line 416-418: "Furthermore, CEUSS has potential value in the diagnostic work-up of SS and could contribute to the assessment of disease severity and progression, since an assessment of the peripheral ductal system and parenchyma is possible in contrast to classic sialendoscopy." - this statement is not supported by the findings. Did the authors mean the central duct system? Moreover, in just 10 cases one cannot assess the severity and progression of a disease. Do we know at all how the contrast agents work in normal glands and how the "normal" distribution is?

Answer: We fully agree with this comment. In this paragraph we speculate about future applications of CEUSS. Many things still need to be studied before such an application can become a reality. We added text in lines 440 – 442. Furthermore, we mention in this paragraph that SGUS (and not CEUSS) could be able to assess disease severity and progression. Here we only suggest that CEUSS offers more contrast than SGUS and that this could help with assessing glandular tissue. We think figure 2 and the videos in the appendices support this suggestion.

Question: Diagnosing and staging of lymphoma is also not possible with CEUSS. It is not shown that CEUSS is less operator dependent than Ultrasound alone.

Answer: We did not claim that diagnosing and staging of lymphoma is possible with CEUSS, but stated that it is impossible using classic sialendoscopy. We apologize for the confusion, caused by this statement. To prevent confusion, we have deleted the lines about diagnosing and staging of lymphoma. For the same reason we have deleted the lines about operator dependency.

Round 2

Reviewer 2 Report

I do understand the authors, that this is just a first study on this subject. Thy authors made all corrections and added comments for the reviewers proposals.
